# Integrative Genomics and Metabolomics Analyses Provide New Insights into the Molecular Basis of Plant Growth Promotion by *Pantoea agglomerans*

**DOI:** 10.3390/microorganisms13092138

**Published:** 2025-09-12

**Authors:** Anna Grazia Ficca, Francesca Luziatelli, Renée Abou Jaoudé, Maurizio Ruzzi

**Affiliations:** Department for Innovation in Biological, Agrofood and Forest Systems (DIBAF), University of Tuscia, Via C. de Lellis, snc, I-01100 Viterbo, Italy; ficca@unitus.it (A.G.F.); f.luziatelli@unitus.it (F.L.); raj@unitus.it (R.A.J.)

**Keywords:** plant growth-promoting rhizobacteria, pan-genome analysis, genome annotation, metabolomics, auxin, *Pantoea agglomerans* strain C1, microbial biostimulants

## Abstract

Plant Growth-Promoting Rhizobacteria (PGPR) are emerging as a sustainable alternative in agriculture due to their environmentally friendly properties and their ability to enhance crop productivity. Among these, *Pantoea agglomerans* has gained attention for its versatility as both a biofertilizer and a biocontrol agent. In this study, we use comparative genomics to gain insight into the genetic diversity and functional specialization of members of this species. The pan-genome analysis of 20 representative *P. agglomerans* strains revealed that 32% of the genes constitute the core genome (2856 out of 8899), while the remaining 68% are classified as accessory or singleton genes, indicating a high level of genomic diversity within the species. Functional annotation showed that core genes are predominantly involved in central metabolic processes, whereas genes associated with specialized metabolic functions are found within the accessory and singleton categories. The comparative analysis demonstrated a mosaic distribution of genes related to nitrogen and sulfur metabolism, heavy metal resistance, defense mechanisms, and oligopeptide uptake, suggesting niche-specific adaptations and metabolic capabilities within this species. Exometabolome profiling of strains associated with different hosts, specifically plant (C1) or human (DSM3493^T^), demonstrated that omics-centered approaches can be utilized to select *P. agglomerans* strains tailored to specific agronomic requirements.

## 1. Introduction

Plant Growth-Promoting Rhizobacteria (PGPR) and microbiome-based technologies have emerged as sustainable alternatives in agriculture due to their eco-friendly nature and ability to enhance crop production [1,2,3].

The use of these systems, either alone or in combination, can have a positive impact on plant and soil health, as well as on holobiont functionality. *Pantoea agglomerans* is a gram-negative bacterium, commonly found in soil, water, and on the surfaces of plants, that has gained significant attention as a versatile PGPR [4,5]. Within the diverse group of beneficial microorganisms, *P. agglomerans* has been proven to be effective as both a biofertilizer and a biocontrol agent [6]. Its various beneficial traits, including phytohormone production, nutrient solubilization, and pathogen suppression, support plant development and increase resistance, making it essential for sustainable agriculture [4,6,7,8,9,10].

The direct and indirect interactions between *P. agglomerans* and plants (mediated by cell–cell contact and exometabolite-induced responses) are essential for the effectiveness of their relationship. This bacterium colonizes the rhizosphere, forming symbiotic associations with plant roots and leaves, and interacts with other microorganisms, stimulating the growth of beneficial PGPR and suppressing pathogens.

In pathogen management, *P. agglomerans* produces antibiotics and siderophores that naturally protect plants by inhibiting harmful bacteria, thereby reducing the need for chemical treatments [11]. For instance, when applied to potato plants, *P. agglomerans* significantly decreased bacterial wilt caused by *Ralstonia solanacearum* [12,13]. Additionally, biosynthetic pathways for lipopeptides and bacteriocins enhance their defense mechanisms. In apple orchards, *P. agglomerans* has been shown to suppress *Erwinia amylovora*, the causative agent of fire blight, demonstrating its potential for biocontrol [14,15].

Beyond pathogen suppression, *P. agglomerans* also plays a significant role in supporting plant growth by producing phytohormones [16,17]. The latter regulate essential growth processes, such as root development, flowering, and fruiting, thereby helping plants to grow more efficiently in diverse environments [18,19]. Additionally, *P. agglomerans* is known for its ability to solubilize essential nutrients in the soil by converting unavailable forms into soluble ones, making these nutrients more accessible to plants, promoting healthier growth and improved crop yields, especially in nutrient-deficient soils [4]. Interestingly, different *P. agglomerans* strains exhibit varying levels of growth promotion and biocontrol capabilities due to their unique functional traits [20,21]. These differences can be studied through the use of microbial genomics and metabolomics, which allow for a deeper understanding of the molecular mechanisms driving these variations.

Genome sequence analysis indicates that several *P. agglomerans* strains possess genes encoding the enzyme for the biosynthesis of phytohormones, such as indole-3-acetic acid (IAA), which enhances root growth and nutrient uptake [22,23]. This analysis revealed that the region surrounding the gene (*ipdC*) encoding indole pyruvate decarboxylase (IPDC, EC 4.1.1.74), the key enzyme of the IPyA pathway, is well conserved among *P. agglomerans* strains, with a nucleotide identity of the 4 kb ORF1-*ipdC*-ORF2 gene cluster ranging from 88 to 100% [22]. In vitro studies carried out with *P. agglomerans* strain C1 revealed that the ability of this strain to produce high levels of IAA depends on the medium composition and cultivation conditions [22,24]. Metabolomics studies demonstrated that, apart from indole auxins, strain C1 excretes several metabolites that can crosstalk with auxins, affect auxin transport or turnover, and modulate the expression of auxin-regulated plant genes, such as *ARF17* [22]. The traits increase plant resilience to environmental stress, thereby improving overall health and productivity [25].

While microbe-based technologies, especially those involving *P. agglomerans*, show promising progress in sustainable agriculture, the lack of strong genetic predictive models based on comparative genome analysis poses notable limitations. Without these models, it is difficult to accurately forecast the full potential and interactions of *P. agglomerans* with different plant species and environmental conditions. This gap can hinder the targeted use of specific strains, potentially reducing their effectiveness in promoting plant growth and fighting pathogens. Additionally, the absence of genetic models limits our ability to predict how various environmental factors may impact the bacterium’s performance, making it more challenging to incorporate it into diverse agricultural systems. Developing more accurate predictive tools is crucial for effectively utilizing *P. agglomerans* and ensuring its consistent application across various farming methods. In this work, we focus on the comparative genome analysis of 20 different *P. agglomerans* strains C1 and correlated this analysis with the metabolomics of taxonomy-related strains (C1 and DSM 3493^T^) to acquire useful information for developing a new approach that integrates genomics and metabolomics for predicting the PGPR and biocontrol traits of specific strains.

## 2. Materials and Methods

### 2.1. Genome Sequences, Bacterial Strains, and Culture Media

The genome sequences of the 20 strains used to construct the pan-genome of *Pantoea agglomerans* were downloaded from the NCBI database (Appendix A). The 9 C1, a *P. agglomerans* strain isolated from leaves of *Lactuca sativa* L. [7] and characterized for its ability to promote the growth of corn [23] and tomato [23,26] and stimulate the rooting of pear shoots [22], was obtained from the collection of the System and Synthetic Microbiology laboratory at DIBAF (University of Tuscia, Viterbo, Italy). *P. agglomerans* DSM3493^T^ was purchased from Leibniz Institute DSMZ-German Collection of Microorganisms and Cell Cultures GmbH (Braunschweig, Germany). The *P. agglomerans* strains were cultivated on LB (Lennox, (Sigma-Aldrich, St. Louis, MO, USA)) medium at 30 °C and stored as glycerol stocks at −80 °C, as described in previous studies [26]. The vegetal peptone–yeast extract (VY) medium used for auxin production is an alternative formulation of LB broth, in which tryptone is substituted with 10 g/L of a GMO-free vegetal peptone derived from peas [27]. When necessary, LB and VY media were supplemented with tryptophan (4 mM) to serve as a precursor and inducer for auxin/indole-3-acetic acid (IAA) biosynthesis.

### 2.2. Comparative and Pan-Genome Analysis

NCBI Taxonomy [28] was used to construct a Newick file containing a set of 118 *P. agglomerans* strains. Subsequently, this file was used to generate a phylogenetic supertree and visualize the position of the twenty strains used in this study. The phylogenetic tree was obtained using the online tool “Interactive Tree of Life” (iTOL ver. 7.0), available at the following URL: https://itol.embl.de (accessed on 13 January 2025).

Comparative genome analysis was performed using the software platform for comparative genomics “Efficient Database framework for comparative Genome Analyses using BLAST score Ratios” (EDGAR, ver.3.2; https://www.uni-giessen.de/de/fbz/fb08/Inst/bioinformatik/software/EDGAR; accessed on 27 January 2025) [29]. The platform was used to calculate the Average Amino Acid Identity (AAI), which evaluates the average amino acid identity values (threshold value of 95–100%) computed by the BLASTp alignment method and, by hierarchical clustering, to generate the corresponding single linkage dendrogram [30].

The EDGAR platform was utilized to determine the pan-genome, core genome, dispensable genome, and singletons of the twenty *P. agglomerans* strains using the BLAST Score Ratio Values (SRVs). For statistical extrapolation, least non-linear squares curve fittings of the observed pan-genome and core genome sizes as functions of the number of analyzed genomes were performed, following the methodologies described by Tettelin et al. [31] and Willenbrock et al. [32].

The functional annotation of the proteins encoded by the pan-genome was conducted using datasets from the Clusters of Orthologous Genes (COG) database [doi.org/10.1093/nar/gkaa1018], which are accessible on the EDGAR platform.

The function based comparison tool provided by the RAST server [33] was used to assess similarities and differences in the presence of functional roles among *P. agglomerans* strains.

The alignments of multiple OppA amino acid sequences were carried out through the ClustalW program (https://www.genome.jp/tools-bin/clustalw, accessed on 27 January 2025). The exported tree file in Newick format was then utilized to generate a phylogenetic tree using the online tool iTOL (iTOL ver. 7.0), available at the following URL: https://itol.embl.de (accessed on 13 January 2025).

### 2.3. In Silico Identification of Genetic Factors Involved in Plant-Bacteria Interactions

The search for genetic factors involved in bacteria-host plant interactions was conducted on the ‘PLaBAse’ platform [34] using the PIFAR-Pred tool. The same platform was employed to annotate the Plant Growth-Promoting Traits (PGPTs) through the PGPT-Pred tool. The PLaBAse service can be accessed at the following URL: https://plabase.cs.uni-tuebingen.de/pb/plabase.php (accessed on 3 March 2025).

### 2.4. Identification of Regulatory Proteins

The TF genes were identified using P2RP (Predicted Prokaryotic Regulatory Proteins) (http://www.p2rp.org; accessed on 10 March 2025) [35], which computes regulatory proteins (RPs) by taking genome DNA contigs as input. All the TF genes detected were categorized into families by P2RP based on domain architecture, according to the scheme implemented in the P2CS and P2TF databases [36,37].

### 2.5. Indole Auxin Production in Shake Flasks

Seed cultures from frozen glycerol stocks of *P. agglomerans* strains were inoculated into Erlenmeyer flasks (500 mL) containing 50 mL of LB broth. The flasks were incubated overnight at 180 rpm and 30 °C until the late exponential phase of growth [optical density at 600 nm (OD_600_) of 4.5]. The seed cultures were used to inoculate 25 mL of tryptophan (4 mM) amended medium at an initial OD_600_ of 0.2. After 24 h of growth at 30 °C, in agitation (180 rpm), 20 mL of each culture was recovered for indole auxins quantification and untargeted metabolomics studies. All experiments were performed in triplicate.

### 2.6. Measurement of Indole-3-Acetic Acid (IAA) and Related Indolic Metabolites Production

Indole auxins excreted into the culture medium were quantified using a colorimetric assay with Salkowski’s reagent, following the method described by Patten and Glick [38]. The cell culture was centrifuged at 15,000 *g* for 10 min, and an aliquot (1 mL) of filter-sterilized supernatant (0.22 μm) was diluted and combined with 2 mL of Salkowski’s reagent (0.5 M FeCl_3_, 35% v/v HClO_4_). The mixture was incubated at room temperature for 20 min, and the absorbance of the resulting pink color was measured at 530 nm. The concentration of indole auxin in the culture was determined using a calibration curve of pure indole-3-acetic acid (Sigma–Aldrich, Milan, Italy) as a standard. All experiments were conducted in triplicate, and the results were expressed in milligrams of IAA equivalent per liter.

### 2.7. Untargeted Metabolomics and Statistical Analysis

Metabolite extraction was performed using an acidified 80% methanol solution, following the protocol established by Paul et al. [39]. A total of 1 mL of samples was extracted using an Ultra-Turrax homogenizer (Ika T-25; Staufen, Germany) and then centrifuged at 1200 rpm. The supernatant was filtered through a 0.22 μm cellulose membrane. The filtered extracts were subjected to untargeted metabolomic analysis performed by oloBion Laboratory (Barcelona, Spain). Metabolite identification and quantification were carried out according to the methodology described by Bonini et al. [40].

Statistical analysis was performed by the one-way analysis of variance using the SigmaStat 3.1 package (Systat Software Inc., San Jose, CA, USA).

## 3. Results

### 3.1. Pan- and Core Genome

To provide a detailed picture of the genetic diversity among *P. agglomerans* strains, a pan-genome analysis was carried out on a set of 20 representative genome sequences retrieved from the NCBI database (Appendix A). The resulting dataset encompassed genomes ranging in size from 4.17 to 5.36 Mbp, with an average length of 4.9 Mbp (Appendix A), dispersed across multiple regions of the supertree constructed using 118 of the 263 available *P. agglomerans* genomes (as of 27 February 2025) (Figure 1).

Pairwise AAI values were calculated for the various genomes and used to display the relatedness among the strains. The analyses showed that the 20 genomes shared AAI higher than 97–99% (Figure 2, left panel). The hierarchical clustering analysis of the AAI data revealed distinct phylogenetic lineages (indicated with A to D in Figure 2, right panel). Lineage A, with most strains (13 out of 20), also includes the PGPR strain *P. agglomerans* strain C1 [26] and the *P. agglomerans* type-strain DSM3493^T^. Lineages B, C, and D had a common ancestor and are paraphyletic to lineage A (Figure 2, right panel). The maximum divergence between *P. agglomerans* genomes was detected against strain DE0421 (Figure 2).

The Edgar 3.2 program was used to analyze the *P. agglomerans* pan-genome in detail (with a 90% BLASTp percentage identity cut-off) and to cluster the genes encoding complete protein sequences into core and dispensable/accessory genomes [41]. The pan-genome exhibited a linear progression, comprising 8899 genes, which is approximately twice the average total number of genes in each of the 20 strains (4435 genes; Figure 3). According to Heap’s Law, as represented by the formula *n* = k × N^−α^ [42], an α ≤ 1 is representative of an open pan-genome, meaning that each added genome will contribute some new genes and the pan-genome will increase, and α > 1 represents a closed pan-genome, in which the addition of new genomes will not significantly affect the pan-genome. We inferred that the pan-genome reaches a maximum at α = 0.72, indicating an open pan-genome.

Within the pan-genome, 2856 genes (present in all 20 strains) were identified as core genes, representing 32% of the entire pan-genome. The dispensable accessory genome comprised 2992 cloud genes, representing approximately 34% of the entire pan-genome. The number of accessory genes occurring in more than 2 genomes varied between 41 (12 genomes) and 679 (2 genomes; Figure 4). The total number of unique genes (singletons) was 3051, about 34% of the entire pan-genome (Figure 4). The number of singletons for each genome varied between 46 (strain C1) and 444 (strain DAPP_PG734; Appendix A).

### 3.2. Functional Annotation of the Pan- and Core Genome Sequences

Functional annotation of all genes of the pan-genome by clusters of orthologous groups (COGs) revealed that functions associated with central metabolism (COG categories “Carbohydrates” and “Amino acid” transport and metabolism), “Transcription”, and “General functions” are over-represented in the core genome (Figure 5; Appendix A). The relative distribution within the COG categories indicated that most of the proteins related to “Translation,” “Coenzyme,” and “Nucleotide transport and metabolism” were encoded from genes of the core genome (Figure 5; Appendix A). Proteins belonging to the COG category “Extracellular structures” were mainly encoded by dispensable genes. Genes encoding “Transposase” were equally distributed between dispensable and singletons, while genes encoding proteins involved in “Transcription” were equally distributed between core and dispensable genomes (Appendix A). About 30% of the genes encoding proteins in the COG categories “Carbohydrates” and “Amino acid” transport and metabolism are part of the dispensable genome. Interestingly, 20% of the proteins in the “Carbohydrates” COG category are encoded by singletons (Appendix A). This suggests that the diversity among *P. agglomerans* strains, which accounts for their presence in various ecological niches, is linked to metabolic differences related to the selectivity of different carbon and nitrogen sources.

### 3.3. Genes Related to Plant Growth-Promotion

We utilized the RAST annotation to identify genetic traits associated with plant growth promotion in the analyzed *P. agglomerans* genomes. The results presented in Appendix A indicate that, with a few exceptions, all strains exhibit numerous traits characteristic of plant growth-promoting rhizobacteria (PGPR).

The operons encoding enzymes involved in auxin (IAA) synthesis through the IPyA (indole-3-pyruvate pathway) and IAM (indole acetamide pathway) pathways are found in all strains. An additional gene encoding an auxin efflux carrier is also conserved across all strains (Appendix A).

The two operons that encode the enzyme responsible for converting arginine into putrescine (*speAB*) and subsequently converting putrescine into spermidine (*speDE*) are present in all genomes.

The genome annotation also revealed several gene clusters involved in mineral phosphate solubilization, including those that encode PQQ-dependent glucose dehydrogenase (*gcd*), membrane-bound gluconate-2-dehydrogenase (*gad*), and a phosphatase-specific transport system (Appendix A). The genome comparison also revealed that all analyzed *P. agglomerans* strains have a duplicated *gad* pathway (*gadCBA_1* and *gadCBA_2* in Appendix A) and several individual genes encoding other P transporters.

Regarding the indirect methods of promoting plant growth, we identified genes that encode enzymes involved in the synthesis of various volatile organic compounds (specifically acetoin and 2,3-butanediol), siderophores, and the EfeUOB transporter (a ferrous iron transporter triggered by low pH and low iron), along with GABA (Appendix A).

It should be noted that the gene cluster, containing the contiguous genes *bdh*, *alsSD*, and *alsR*, which encode enzymes involved in the biosynthesis of acetoin (3-hydroxybutanone) and 2,3-butanediol, was not found in strains TH81 and L15 (Appendix A). Similarly, the operons encoding the ferrous iron transport proteins A and B (*feoBA* operon) and the key enzyme for GABA biosynthesis (*gabRTP* operon) were also absent in these strains (Appendix A). Additionally, the operon for GABA biosynthesis and the gene cluster (*fepACGDB-entFSCEBA-fes-ybdZ*) responsible for the biosynthesis and transport of enterobactin siderophores were not present in strain DE0421 (Appendix A).

### 3.4. Nitrogen Metabolism

Functional annotation of the 20 genomes revealed the presence of 6 operons containing genes related to nitrogen metabolism: the nitrate extracellular transporter (*nrtABC* gene cluster and the gene encoding the response regulator NasT); the nitrate/nitrite response regulator NarL (*narQP*); the dissimilatory nitrate reduction pathway (*nirBD* and *narGHIJ* gene clusters and the nitrate/nitrite transporter NarU); the assimilatory nitrate reduction pathway (*nasBC* gene cluster); the response to nitrosative stress (*norRVW* gene cluster and the gene encoding the nitrite-sensitive transcriptional repressor NsrR); and the ammonia assimilation pathway (Appendix A). Notably, all these gene clusters were present only in 3 strains: C1, RIT273, and DOAB1048 (Appendix A).

### 3.5. Sulfur Metabolism

RAST annotation revealed genes encoding proteins involved in 7 sulfur metabolic pathways (Appendix A). The complete set of genes necessary for sulfate uptake and reduction to sulfide and sulfite, cysteine metabolism, and alkanesulfonate assimilation was present in all the *P. agglomerans* genomes analyzed in this study (Appendix A). The gene encoding the ATPase component of the ABC-type nitrate/sulfonate/bicarbonate transport system was absent in most of the 20 strains (Appendix A).

### 3.6. Tolerance Against Heavy Metals

Genes associated with heavy metal tolerance display variability across the twenty genomes analyzed (see Appendix A). All genes involved in arsenic tolerance were exclusively identified in the C1 genome. Conversely, gene clusters related to copper and cadmium tolerance were detected solely in the C1 and LMAE2 strains (refer to Appendix A).

The *arsRH* operon was identified in 6 out of 20 genomes, whereas the *arsRBC* gene cluster was present in 10 out of 20 genomes. Additionally, in three genomes, the arsenate reductase (glutaredoxin) *arsC* gene was absent (Appendix A).

### 3.7. DNA Phosphorothioate (PT) Modification

DNA phosphorothioate (PT) modification is one of the defense mechanisms that bacteria have evolved against phages. Among the 20 *P. agglomerans* genomes we analyzed, only the C1 genome contains genes associated with the *dnd-dpt operon,* which encodes the proteins (DND) involved in DNA PT (Appendix A). The C1 genomic region, encompassing the two gene clusters, is approximately 13,300 bp long and includes a putative integrase gene located near a tRNA-Leu gene at its 5′ end. In contrast, a transposase gene is found at the 3′ end (Appendix A).

### 3.8. The Oligopeptide (opp) Gene Cluster

RAST annotation revealed the presence of a specialized ATP-binding cassette (ABC) transporter composed of membrane-associated proteins known as “Opp proteins,” which are essential for the uptake of oligopeptides in bacteria. The genes encoding the five proteins that constitute the oligopeptide transport system are consistently arranged in an operon (*oppABCDF*). We identified an opp gene cluster comprising two copies of the *oppA* gene (*oppA1* and *oppA2*; Appendix A, line A). Although *oppA1* and *oppA2* are similar, they are not identical; both genes are present in all 20 *P. agglomerans* genomes and are clustered separately into groups A1 and A2 in the phylogenetic tree shown in Appendix A. Six of the 20 strains (C1, 190, 4, RT273, RIT710, and JM1; designated as group A3 in the phylogenetic tree shown in Appendix A) contained an additional *oppABCDF* gene cluster featuring a single copy of the *oppA* gene (*oppA3*, Appendix A, line B). We utilized the *oppA3BCDF* gene cluster from strain C1 for BLAST analysis to locate a similar opp system in other *P. agglomerans* genomes. This analysis identified a homologous gene cluster associated with a plasmid-borne IS30-family transposase (Appendix A, line C).

### 3.9. Whole-Genome Comparison Between P. agglomerans Strains C1 and DSM3493^T^

We then compare the genomes of our C1 isolate, which has been shown to enhance the growth of several food crops [16,22,23,26], with the *P. agglomerans* type strain DSM 3493^T^ [42]. The hierarchical cluster analysis indicated that, although the 2 genomes share over 99.5% AAI, these strains are phylogenetically separated into two distinct clades within group A (Figure 2). Utilizing the progressive Mauve software (ver. 2.4.0), the genome alignment revealed 37 collinear blocks and several regions exhibiting inversion and rearrangement (Appendix A). These variations in chromosomal architecture can impact the transcriptional regulation of conserved genes.

Comparative analysis of coding sequences revealed that the two genomes share 3929 genes (core genes, Appendix A), corresponding to 89.7% and 93.5% of the C1 and DSM3493^T^ gene content, respectively. We also identified 692 unique genes that were differentially distributed between the two genomes: 407 in strain C1 and 285 in strain DSM3493^T^.

### 3.10. Genetic Traits Associated with Bacteria-Plant Interaction and Plant Growth Promotion

To gain further insights into the differences between the two strains, we analyzed their genomes using PlaBase’s predictive tools. This web resource provides two annotated databases (Figure 6): one for predicting functions related to bacterial-plant interactions (utilized with the PIFAR-Pred tool) and the other for traits associated with plant growth promotion (employed with the PGPT-Pred tool).

The PIFAR-Pred identified over 260 proteins (263 in DSM3493^T^ and 265 in strain C1) that could be important in bacterial-plant interactions. About half of these proteins were associated with three categories: exopolysaccharides (EPS), hormones (IAA, cytokinins, and salicylic acid), and detoxification (isothiocyanate; Figure 6, panel A).

Differences between the two strains were primarily related to the number of genes encoding proteins involved in the biosynthesis of IAA (15 proteins, representing 7% of the total in strain C1, and 14 proteins, representing 6% of the total in DSM3493^T^), amylovoran (one additional protein in strain C1, 65 versus 64, corresponding to 25% of the total proteins detected by PIFAR-Pred), and proteins involved in cell adhesion (25 proteins in DSM3493^T^ versus 21 in strain C1, corresponding to 10% and 8% of the total, respectively; Figure 6, panel A).

PGPT-Pred identified several proteins correlated with direct (28% of the total in strain C1 and 27% in DSM3493^T^) and indirect (72%) plant growth promotion. The direct effects refer to biofertilization, plant signal production, and bioremediation. The indirect effects include plant tissue colonization, competitive exclusion of potential pathogens, and amelioration of plant stress (Figure 6, panel B).

In this analysis, differences between the two strains were primarily related to the number of genes associated with heavy metal detoxification, which were more abundant in strain C1 (as also reported in Appendix A), and genes related to surface attachment, which were more abundant in strain DSM3493^T^, comprising 3% of the total (Figure 6, panel B).

### 3.11. Type VI Secretion System

The type VI secretion system, T6SS, is a complex multi-component secretion machinery used for interbacterial interactions, ranging from commensal collaboration to competition. Possession of a T6SS can confer a significant ecological advantage to bacterial cells. Through sequence comparison, we examined the genomes of the two strains to identify homologs of the T6SS genes. We identified gene clusters that encode T6SS components by bioinformatics analysis: one of 75 Kbp in DSM3493^T^ (Cluster 1) and two (one of 94 Kbp [Cluster 1] and the other of 31 Kbp [Cluster 2]) in C1. Both clusters contain, at the 3’-end, an inverted duplication that generates an extra copy of six genes (Figure 7). No gene duplication was observed in Cluster 2 (strain C1).

### 3.12. Regulatory Proteins (RP)

Identifying the RP repertoire is crucial for assessing cell regulatory networks. Genome annotation using the RP prediction platform P2RP identified 393 genes in strain C1 and 375 in strain DSM3493^T^, which encode two-component systems, transcription factors, and other DNA-binding proteins (Figure 7). These RPs were classified into 52 families (Figure 8), representing 8.9% of the theoretical proteomes in both strains.

### 3.13. Auxin Production Under Different Culture Conditions

To evaluate how the genetic differences between *P. agglomerans* strain C1 and DSM3493^T^ affect their ability to produce specific metabolites, we focus on auxin/IAA biosynthesis, a trait conserved within the species (Appendix A). Specifically, we investigated whether the medium composition, which we previously showed influences auxin production in strain C1 [24], has a similar effect on strain DSM3493^T^. The strains were cultured on 2 different media that varied in their organic nitrogen sources (peptone derived from animal or plant sources). The culture media were supplemented with tryptophan (4 mM), a precursor and inducer of auxin/IAA biosynthesis, and the experiments were conducted in shaken flasks under optimized shaking and medium volume-to-flask volume conditions [24].

Figure 9 shows that the comparison of auxin accumulation in the culture medium revealed significant differences in the volumetric yield of auxin produced by the two strains. Data obtained after 18 h of incubation indicated that strain C1 exhibited the highest levels of auxins in both media. No further increase in the IAA level was observed by increasing the incubation until 48 h, in agreement with our previous observations with strain C1 [24]. The highest concentrations for both strains were found in the VY medium with vegetable peptone as the carbon and nitrogen source, measuring 123.3 ± 3.6 mg IAA_equ_ L^−1^ for strain C1 and 76.3 ± 1.4 mg IAA_equ_ L^−1^ for strain DSM3493^T^ (Figure 9). The relative increase in auxin levels when replacing tryptone with pea peptone was more significant in DSM3493^T^ than in the C1 strain, demonstrating a 25% increase compared to a 14% increase.

### 3.14. Comparative Untargeted Metabolomics

To better understand the metabolic differences between strain C1 and DSM3493^T^, we analyzed the excreted metabolites in LB medium supplemented with tryptophan using an untargeted metabolomics approach. UHPLC-ESI-Q-TOF-MS analysis of the exhausted growth medium revealed a total of 151 shared features. Of these, 17 metabolites showed significant changes (*p* ≤ 0.05), with at least a two-fold difference in their relative abundance: 6 were more than two times higher in the C1 exometabolome, and 11 were more than two times more abundant in the exometabolome of DSM3493^T^ (Table 1). These metabolites belong to 14 distinct classes. Regarding the dipeptides class, which encompasses multiple metabolites, Tyr-Pro was 2.8-fold more abundant in the C1 exometabolome, and 5-oxoproline concentration was 3.1-fold higher in the DSM3493^T^ exometabolome, indicating that these metabolites were differentially enriched across the two *P. agglomerans* strains (Table 1). Similarly, within the indole class, tryptophol was found to be 38 times more abundant in the C1 exometabolome, whereas 6-methoxytryptamine was 11 times more prevalent in the exometabolome of DSM3493^T^ (Table 1).

## 4. Discussion

The comprehensive comparative genomic analysis of *P. agglomerans* provides new insights into the genetic diversity and functional specialization of members of this taxon for their plant growth-promoting and biocontrol traits. The study emphasizes the significance of gene variability among strains, particularly in accessory and unique genomic regions, in enabling them to colonize diverse ecological niches and provide benefits to plants.

The pan- and core genome structures highlight the species’ adaptability. Analyzing 20 representative *P. agglomerans* strains revealed an open pan-genome with an α of 0.72, consisting of 8899 genes—approximately double the typical genome size of about 4435 genes. Only 32% of these genes form the core genome, while the remaining 68% are accessory or singletons, indicating high genomic diversity within the species. These results support earlier studies on PGPRs, which also demonstrate open pangenomes, pointing to active horizontal gene transfer and adaptive evolution [41].

The high average amino acid identity (AAI > 97%) confirms that the species are consistent, but hierarchical clustering based on AAI identified four distinct lineages (A–D), indicating evolutionary divergence (Figure 2). Notably, lineage A contained both the PGPR strain C1 and the type strain DSM3493^T^, whereas they were placed in separate subclades (Figure 2, right panel).

Metabolic versatility and plant interaction genes facilitate niche adaptation, thereby playing a crucial role in shaping plant-microbe interactions. Functional annotation of selected *P. agglomerans* genomes revealed that core genes are predominantly involved in central metabolic processes, such as amino acid and carbohydrate metabolism, satisfying fundamental physiological requirements. Conversely, genes linked to extracellular structures, transposases, and specialized metabolic functions are primarily located within the accessory and singleton categories, implying their participation in ecological diversification. COG analysis revealed a significant enrichment of carbohydrate metabolism genes in accessory and singleton genomes (up to 20%; Appendix A), suggesting a level of metabolic flexibility that likely enables individual strains to adapt to specific carbon sources in the rhizosphere.

The distribution of PGPR-related genes across all strains highlights their ecological functions. Genes for IAA biosynthesis (through IAM and IPyA pathways), spermidine production (*speAB* and *speDE*), phosphate solubilization (*gcd* and *gad*), and siderophore synthesis were commonly found in the analyzed genomes (Appendix A).

Nitrogen and sulfur metabolism act as adaptations specific to certain niches [43]. Interestingly, genes related to nitrogen processes—such as assimilatory and dissimilatory nitrate reduction, nitrosative stress response, and ammonia assimilation [44]—were found only in strains C1, RIT273, and DOAB1048 (Appendix A). Interestingly, all three strains were isolated from aerial parts of the plant in contexts in which the nitrogen content was altered: C1 was isolated from the phyllosphere of lettuce (*Lactuca sativa* L.) plants treated with vegetal-derived protein hydrolysates [45]; RIT273 from shrubs willow (*Salix* spp.) [46], a tree species able to grow in nutrient-poor environments [47]; DOAB1048 from wheat (*Triticum aestivum* L.) leaf necrotic tissues caused by *Xanthomonas translucens* [48], a phytopathogen whose interaction with the plant is slowed down by high levels of nitrogen [49].

Notably, in the type strain DSM3493^T^, the genes related to nitrate transport, assimilatory nitrate reduction, and nitrosative stress were missing (Appendix A). Their absence in DSM3493^T^ and other strains indicates these metabolic traits are niche-specific, possibly offering benefits in nitrogen-poor soils.

The nitrosative stress pathway [50] is identified in only 3 of the 20 genomes analyzed; however, all strains carry the gene encoding the nitrite-sensitive transcriptional repressor NsrR [51]. In the DE0421 strain, genes encoding the nitrite reductase (*nirB* and *nirD*) and the nitrate/nitrite response regulator NirP were not detected (Appendix A). The ecological significance of these genetic variations warrants further investigation, especially in relation to the capacity of different strains to promote plant growth.

The comparative analysis of *P. agglomerans* genomes further revealed several genes associated with sulfur metabolism, indicating niche-specific adaptations and metabolic capabilities within this species (Appendix A). All examined genomes of *P. agglomerans* possessed the complete set of genes necessary for sulfate uptake and its subsequent reduction to sulfide and sulfite, cysteine metabolism, and alkanesulfonate assimilation (Appendix A). The presence of several sulfur metabolism genes underscores the metabolic diversity and specialization employed by various *P. agglomerans* strains to acquire and utilize sulfur-containing compounds across different environments [52].

A complete set of genes essential for the uptake and conversion of taurine, an organic sulfur source, into sulfite was also identified, indicating that *P. agglomerans* strains can utilize both organic and inorganic sulfur sources as an adaptive response to environments with limited nutrients (Appendix A). Additionally, the discovery that the gene encoding the ATPase component of the ABC-type nitrate/sulfonate/bicarbonate transport system was only present in the DAPP-PG734 genome suggests a unique adaptation for nutrient uptake in this strain (Appendix A).

The presence in *P. agglomerans* of genes associated with different pathways for sulfur metabolism underscores the genomic adaptability inherent to this species. It also emphasizes the potential for utilizing these strains in biotechnological applications, such as bioremediation strategies for environments contaminated with sulfur. Further research is required to elucidate the specific mechanisms governing sulfur metabolism in *P. agglomerans* and to investigate the ecological and evolutionary factors that contribute to the observed genomic diversity [53].

The comparative genome analysis also revealed that the distribution of genes associated with heavy metal tolerance exhibits significant variation across the 20 genomes examined (Appendix A). These strains have developed distinct adaptive strategies to manage heavy metal stress [54]. Specific gene clusters related to arsenic, copper, and cadmium tolerance were identified solely in the C1 genome (Appendix A), underscoring its unique genetic composition and potentially enhanced capacity to survive in highly contaminated environments [55]. The occurrence of the *arsRH* operon in 6 out of 20 genomes and the *arsRBC* gene cluster in 10 out of 20 genomes (Appendix A) indicates that arsenic resistance is a relatively common trait; however, it is not universal, thereby reflecting diverse environmental pressures and evolutionary trajectories [56]. Notably, in strains JM1, B3, and DOAB1048, the *arsRBC* operon is truncated, lacking the arsenate reductase (glutaredoxin) *arsC* gene (Appendix A), which suggests a potentially diminished or alternative arsenic resistance mechanism, possibly impairing the capacity of these strains to reduce arsenate [57].

The C1 and LMAE2 strains were the only ones harboring gene clusters associated with copper and cadmium tolerance (Appendix A), suggesting that these strains have undergone specific adaptive events, possibly driven by exposure to elevated levels of these metals [58]. This differential distribution underscores the mosaic nature of heavy metal tolerance genes and the diverse evolutionary paths taken by different strains in response to heavy metal contamination. These strain-specific traits, coupled with variations in auxin production and metabolic output, further underscore the diverse ecological adaptations within this group of organisms. The observed genomic flexibility, facilitated by horizontal gene transfer [26] and adaptive evolution, allows for rapid adaptation to changing environmental conditions.

Data reported in Appendix A also indicated that C1 is the only *P. agglomerans* strain analyzed in this study that has a complete DNA phosphorothioate (PT) modification system: one of the defense mechanisms that bacteria have evolved against foreign DNA and shield themselves from bacteriophage attacks [59]. This system differentiates self- and non-self-DNA through *dnd*-mediated DNA PT modification, which substitutes the non-bridging oxygen in the DNA sugar-phosphate backbone with sulfur [60]. This alteration provides nuclease resistance at the PT linkage and may enhance its genomic stability and ecological fitness [59].

The complex mechanisms that regulate nutrient uptake in bacteria are essential for their survival and adaptability to various environments [61]. Among these mechanisms, ATP-binding cassette (ABC) transporters, especially the transport systems that facilitate the import of oligopeptides, play a particularly significant role [62].

The presence in *P. agglomerans* genomes of a specialized ABC-type transporter characterized by membrane-associated proteins known as Opp proteins highlights the importance of oligopeptide uptake for this species [63]. These Opp proteins, which are integral components of the oligopeptide transport system, are encoded by the *oppABCDF* gene cluster [64].

Our analysis of the *opp* gene cluster in *P. agglomerans* revealed a complex genomic landscape characterized by gene duplication, horizontal gene transfer, and phylogenetic divergence (Appendix A). We identified an *opp* gene cluster containing two copies of the *oppA* gene, designated *oppA1* and *oppA2*. These copies are similar, yet not identical, across all 20 *P. agglomerans* genomes examined (Appendix A).

The presence of 2 distinct *oppA* paralogs suggests potential subfunctionalization or neofunctionalization, allowing the bacterium to adapt to a broader spectrum of oligopeptides and environmental conditions. Phylogenetic analysis further categorized these *oppA* genes into two separate groups, A1 and A2 (Appendix A), indicating their evolutionary divergence.

Moreover, it was observed that 6 of the 20 strains possess an additional *oppABCDF* gene cluster, which includes a single copy of *oppA* (*oppA3*, Appendix A). This observation suggests that gene duplication has occurred, followed by sequence diversification. The identification of a homologous gene cluster associated with a plasmid-borne IS30-family transposase highlights the role of horizontal gene transfer in shaping the genome of *P. agglomerans*, potentially facilitating the acquisition of novel metabolic capabilities [65].

In conclusion, the pan-genome analysis demonstrates that the strain possesses an increased repertoire of genes that facilitate plant-beneficial traits, augment stress resilience, and bolster microbial competitiveness, thus rendering it a promising candidate for sustainable agricultural applications.

To better analyze the peculiarity of the strain C1, we directly compared its genome with that of the type strain DSM3493^T^. Although their genomes are very similar (AAI > 99.5%), they show significant differences in gene content and regulation (Appendix A). The Mauve alignment of their genomes revealed extensive genomic rearrangements, including inversions and translocations in conserved regions that can be associated with different phenotypes (Appendix A).

Strain C1 also features a more elaborate Type VI secretion system (T6SS) comprising two genomic clusters with duplicated effector regions and orphan toxin-immunity pairs (Figure 7). Cluster 1 begins with the region conserved across both strains, encompassing the genes *tssJ* to *tssC*, which encode the core proteins of the T6SS. Following this conserved segment is the “*hcp* island,” located between the *hcp* and *tagH* genes, which consists of a series of genes encoding four “orphan immunity proteins,” including a member of the Tai4 family of immunity proteins [66] and proteins featuring the lysozyme inhibitor LprI domain [67]. 

In the C1 genome, we observed an effector/immunity gene pair, *pse2*/*psi2*, located outside the “*hcp* island”; Cluster 1 of DSM3493^T^ lacks the *psi2* gene.

The sequence analysis also revealed the presence of a second region, conserved in both strains, that extends from the *tagH* gene to the *tagE* gene, known as accessory components (Tag for Type VI accessory genes). These genes are involved in modulating the assembly of the system and/or contributing to its regulation.

Following this conserved area was a highly dynamic region that encodes several effector and immunity pairs: the “*vgrG*” and “*paar*” islands. The C1 islands contain two copies of the gene encoding the specialized glucosaminidase immunity VgrG protein (*vgrG1* and *vgrG2*), two copies of the gene for PAAR domain-containing proteins (*paar1* and *paar2*), one copy of the *egg* gene, which encodes DUF1795 or Eag (a DcrB-domain-containing protein that acts as an adaptor for T6SS specialized effectors), and an additional gene that encodes a DUF2169-domain-containing protein (indicated as gene 28 in Figure 7), which may function as a T6SS adaptor and chaperone [68,69]. This configuration implies that strain C1 may possess an augmented capability for interbacterial competition or the regulation of symbiotic relationships [69].

In contrast, DSM3493^T^ contains a “*vgrG* island” with one copy of the *vgI* gene and a “*paar* island” with one copy of the *egg* gene.

The comparative analysis of the two genomes also indicated that the number of genes encoding regulatory proteins (RP) varied between the two strains: 393 in C1 and 375 in DSM3493^T^. These RPs were classified into 52 families, representing 8.9% of the theoretical proteomes in both strains. For each strain, the most predicted transcription factor (TF) family was LysR, the most common type of transcriptional regulator in the prokaryotic kingdom, responsible for regulating amino acid biosynthesis, along with Two-Component Systems (His-Kinase). A total of 11 families were found to be barely present in DSM3493^T^ compared to their C1 counterparts. The C1 genome showed additional regulatory genes: five encoding Xre-type regulatory factors; two for AraC regulators (involved in modulating carbon metabolism, cell wall synthesis, stress responses, and pathogenesis regulation); three for ArsR (which regulates arsenic resistance) and OmpR (responsible for biosynthesis of membrane components); along with two extra copies of LysR. In DSM3497^T^, we identified two additional copies of TetR and several genes related to antibiotic resistance responses. The latter might be related to the fact that DSM3497^T^ was isolated from a knee laceration [70].

The comparative analysis of C1 and DSM3493^T^, utilizing PlaBase’s predictive tools, also disclosed differences in strain-specific traits associated with bacterial-plant interactions and plant growth promotion. The PIFAR-Pred analysis identified 263 proteins in DSM3493^T^ and 265 proteins in C1 that are potentially implicated in bacterial-plant interactions, suggesting conserved features among various strains of *P. agglomerans*. Many of these proteins were linked to exopolysaccharides, hormones, and detoxification processes, particularly isothiocyanates, thereby illuminating the diverse mechanisms through which these bacteria can influence plant physiology and defense responses [26,71]. Exopolysaccharides are recognized for their role in promoting plant growth, although they are less extensively studied [72]. Hormone biosynthesis, encompassing auxins such as IAA, cytokinins, and salicylic acid, enables bacteria to impact plant growth and development directly [73]. Detoxification strategies, notably involving isothiocyanates, demonstrate the strains’ capacity to mitigate plant stress induced by environmental toxins or pathogens. The PGPT-Pred analysis further categorized proteins based on their functions in the direct and indirect promotion of plant growth, providing a comprehensive overview of these bacteria’s beneficial activities [74]. Direct effects, accounting for 28% of the total, were associated with biofertilization, production of plant signaling molecules, and bioremediation, underscoring the ability of these strains to enhance nutrient availability and improve environmental conditions for plants directly. Indirect effects, constituting 72%, included plant tissue colonization, competitive exclusion of potential pathogens, and reduction in plant stress, thereby highlighting the vital role of these bacteria in safeguarding plants against both biotic and abiotic stressors.

Phenotypic assessments demonstrated that C1 consistently yielded higher levels of indole-3-acetic acid (IAA) than DSM3493^T^ under various cultivation conditions, with the maximum concentrations observed in media supplemented with vegetable-derived peptone. Although both strains exhibited responses to alterations in medium composition, the increase in IAA was more pronounced in DSM3493^T^ (25%) relative to C1 (14%). This variation may be attributed to differences in the regulation of IAA biosynthetic genes, such as *amiE* and *ipdC*, which are conserved across all genomes (Appendix A).

Untargeted metabolomics further confirmed strain-specific metabolic profiles. Seventeen metabolites exhibited significant differences between C1 and DSM3493^T^ when grown on a nutritionally rich peptone medium, highlighting variations in the composition of the exometabolome (Table 1). These metabolites included tryptophan derivatives belonging to the indole and indoleacetic acid classes, such as tryptophol (which was 38-fold more abundant in C1) and indole-3-acetic acid (IAA), which was 3-fold higher in the C1 exometabolome compared to DSM3493^T^) [Appendix A]. Both compounds, particularly IAA, are auxin-like phytohormones that regulate plant development by stimulating the formation of lateral roots and increasing root length [75,76]. Tryptophol (indole-3-ethanol) can be converted to IAA and also exhibits antifungal activity, which means it can be a storage form of auxin precursors (indole-3-acetaldehyde) and a biocontrol agent. The simultaneous production of elevated levels of IAA and tryptophol is a characteristic that C1 and other *P. agglomerans* PGPR strains have developed to foster a mutually beneficial relationship with host plants. In contrast, the indole-related compound enriched in the DSM3493^T^ exometabolome was 6-methoxytryptamine (11-fold higher than in C1); this tryptamine derivative can act as a serotonin receptor agonist and may be a trait specific to a pathogenic strain that has coevolved with its human host. Another tryptophan derivative whose abundance significantly increased (*p* ≤ 0.05) in the C1 exometabolome was indole-3-acetamide; however, the increase in this IAA precursor was lower than 2-fold (+1.6-fold; Appendix A).

Dipeptides, such as Tyr-Pro, flavones like 2′,3′-dimethoxy-3-hydroxyflavone, and azoles such as 4-methyl-5-thiazoleethanol (a vitamin B1 precursor), which were significantly more abundant in C1, are all compounds known for their role in promoting the establishment of a mutualistic association between PGPR and the host plant.

A last metabolite that distinctly differentiates the exometabolome of C1 and DSM3493^T^ is fructose, which exhibits a 5.7-fold higher abundance in C1. The mechanism of sugar transport constitutes one of the genomic characteristics that distinguish plant-associated bacterial strains from non-plant-associated counterparts [77]. Moreover, the augmentation of soluble sugar levels facilitated by PGPR can bolster plant growth, particularly under stress conditions, and enhance plant resistance [78].

## 5. Conclusions

The present study demonstrates that omics technologies provide a robust foundation for the functional analysis of *P. agglomerans*. The genomic and functional diversity observed among *P. agglomerans* strains underscores the necessity of employing omics-centered tools, including comparative genomics, metabolomics, and predictive annotation software such as PIFAR-Pred and PGPT-Pred, to identify traits associated with the host (human or plant). These techniques facilitate the identification of lineage-specific features and highlight the importance of strain-level resolution in the evaluation of PGPR. Furthermore, they validate the use of genome-informed approaches for selecting microbial strains tailored to specific agronomic requirements. In conclusion, the comparative analyses further demonstrate that strain C1, in particular, exhibits an enhanced genetic repertoire for plant-beneficial traits, stress tolerance, and microbial competition, making it a strong candidate for sustainable agricultural applications. Furthermore, our results show that the integrated application of comparative genomics and metabolomics can be utilized to accurately predict the potential biotechnological applications of newly isolated strains. The multi-omics strategy used in this work provides a robust framework for elucidating the functional potential of microbial genomes while offering a detailed chemical profile of secondary metabolites. These insights are crucial for advancing microbial biotechnology, particularly in the identification of novel PGPR strains or bioactive compounds and the development of sustainable biotechnological processes.

## Figures and Tables

**Figure 1 microorganisms-13-02138-f001:**
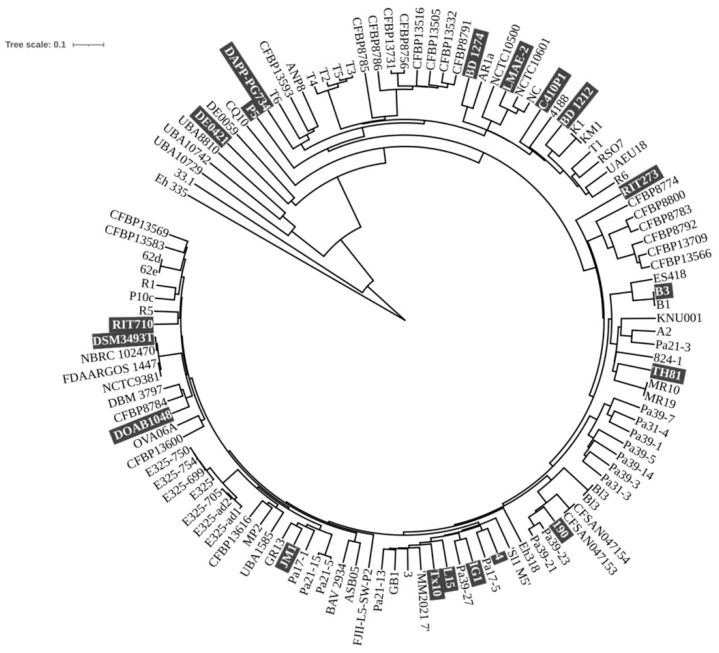
Phylogenetic tree of 117 *P. agglomerans* strains based on conserved genes comparison. The highlighted text indicates the strains utilized for comparative genomics. The C1 strain is not reported in the picture derived from the NCBI site. The genome sequences were retrieved from the NCBI database (https://www.ncbi.nlm.nih.gov/data-hub/genome/; accessed on 1 October 2024).

**Figure 2 microorganisms-13-02138-f002:**
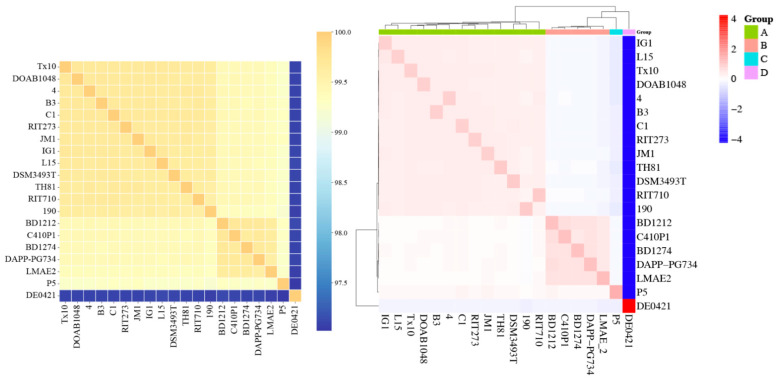
A matrix indicating the pairwise percentage average amino acid identities of the twenty *P. agglomerans* strains. Colors in the heatmap represent bands of percent identity; genome pairs sharing higher/lower amino acid identity are shaded more darkly (dark yellow to dark blue; left panel). The degree of similarity between the genomes was used for hierarchical cluster analysis, resulting in the single linkage dendrogram plot displayed in the right panel. Four distinct clades (A to D) within this species, which can be differentiated through hierarchical clustering, are highlighted in color in the right panel. The values were calculated using the EDGAR 3.2 platform [see Section 2].

**Figure 3 microorganisms-13-02138-f003:**
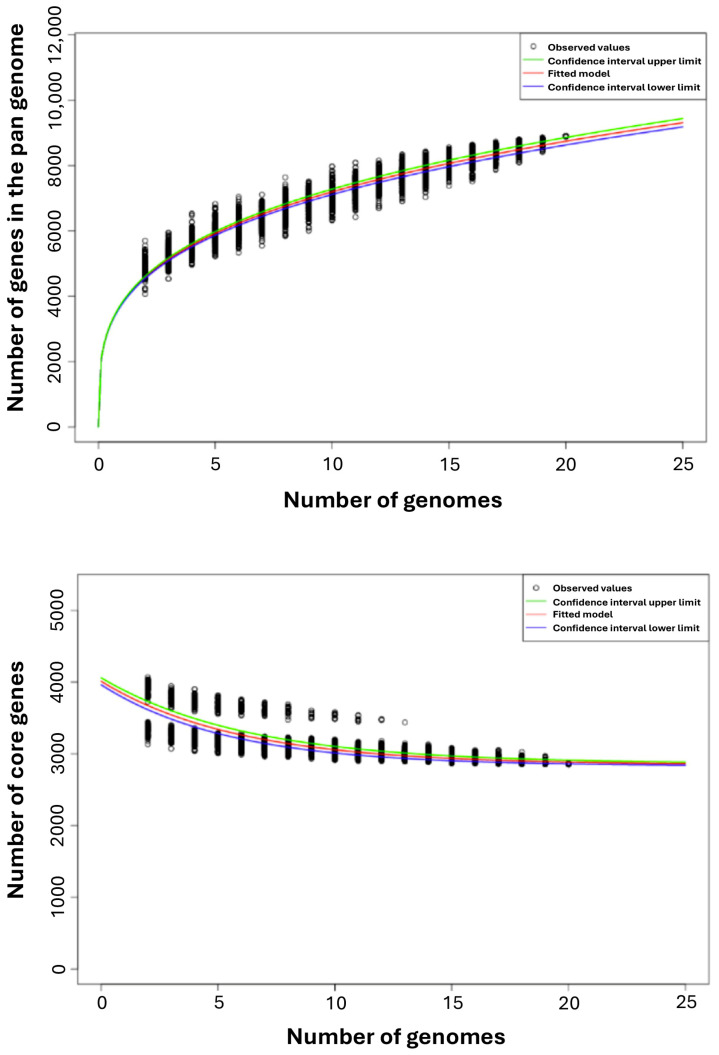
Theoretical estimation of the exponential decay model based on the pan- and core genome sizes. (**Upper** panel) Estimation of pan-genome size based on the Tettelin model fit OMCL clusters [31]. (**Lower** panel) Estimation of core genome size based on the Willenbrock model fit to OMCL clusters [32].

**Figure 4 microorganisms-13-02138-f004:**
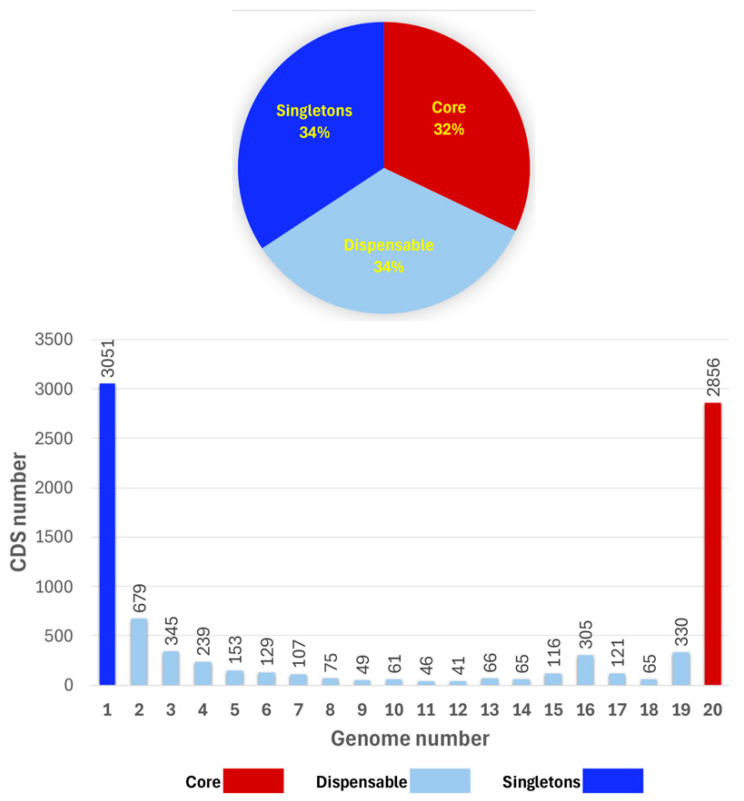
The pan-genome analysis of the 20 *P. agglomerans* genomes reveals the distribution of genes as a function of the number of genomes analyzed. The core genome comprises the genes present in all genomes, while singletons refer to genes that occur in only one genome. The dispensable genome includes genes shared by 2 to 19 genomes.

**Figure 5 microorganisms-13-02138-f005:**
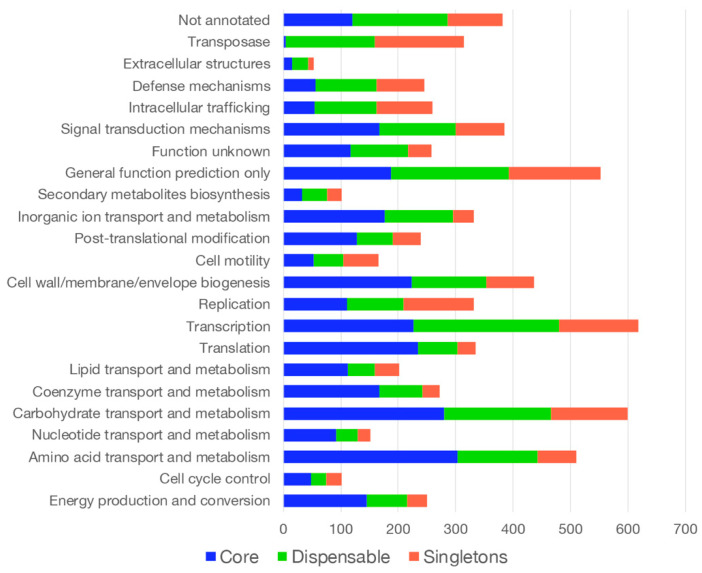
A cluster of orthologous groups (COG) classification of putative proteins encoded by the pan-genome components: core, dispensable, and singleton genes. The analysis was carried out on the EDGAR 3.2 platform [see Section 2].

**Figure 6 microorganisms-13-02138-f006:**
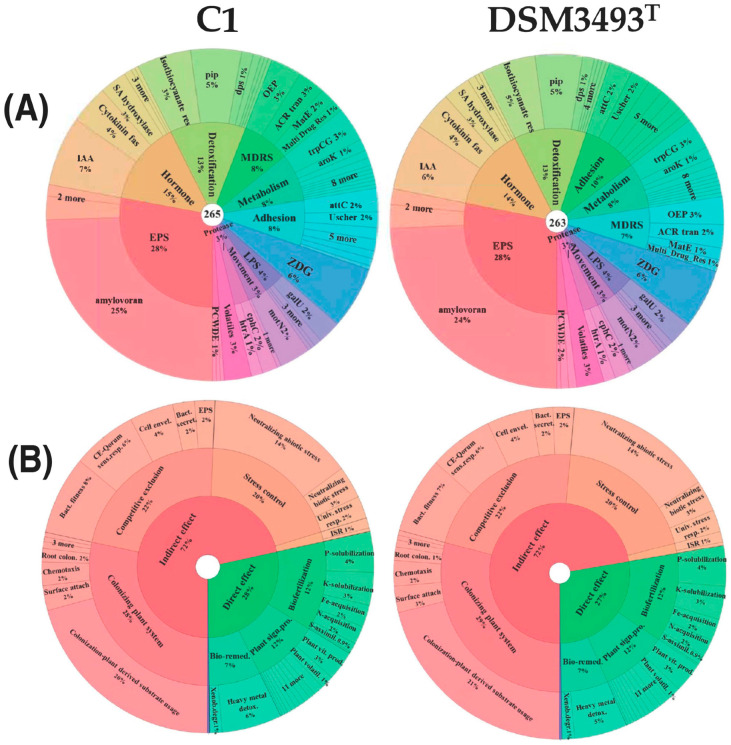
Krona plot representation of the major plant growth-promoting traits found in *P. agglomerans* strain C1 and DSM3493T. Panel (**A**) Genes annotated by PIFAR-Pred. Depth of annotation is shown to level 2 of 3, excluding the accession numbers. EPS, exopolysaccharide; IAA, indole-3-acetic acid; MDR, multidrug resistance; OEP, outer membrane efflux proteins; LPS, lipopolysaccharide; PCWDE, plant cell wall-degrading enzyme; ZDG, zeaxanthin diglucoside. Panel (**B**) genes annotated by PGPT-Pred. Identification of PGPTs was performed by BLASTp and HMMER annotation against the PGPT-BASE. Depth of annotation is shown to level three of six, excluding pathways, gene names, and accession numbers.

**Figure 7 microorganisms-13-02138-f007:**
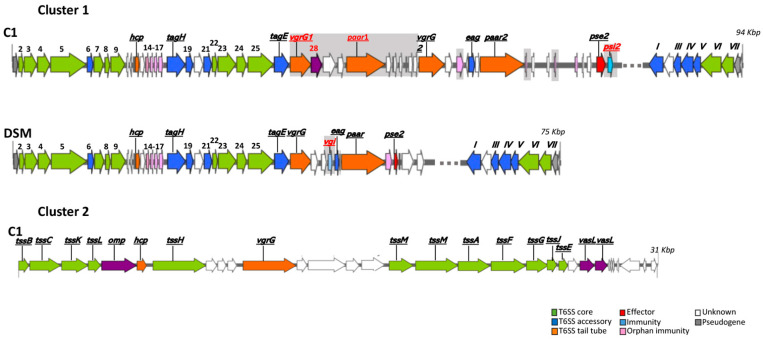
Comparison of the T6SS-6 loci in *P. agglomerans* strain C1 and DSM3497^T^. Blocks of related genes are colored the same. Genes without related or homologous genes are shaded gray. Some genes are indicated with sequential numbers: 2-*tssJ*; 3-*tssK*; 4-*tssL*; 5-*tssM*; 6-*tagF*; 7-*tssA*; 8-*tssB*; 9-*tssC*; 19-*tagG*; 21-*tagJ*; 22-*tssE*; 23-*tssF*; 24-*tssG*; 25-*tssH*; 26-*tagE*; 28-DUF-2169 domain containing protein; I-*tagE2*; III-*tagG2*; IV-tagH2; V-*tagF2*; VI-*tssM2*; VI-*tTssL2*; *omp*; outer membrane protein; *vasL*; and Type VI secretion-related protein VasL.

**Figure 8 microorganisms-13-02138-f008:**
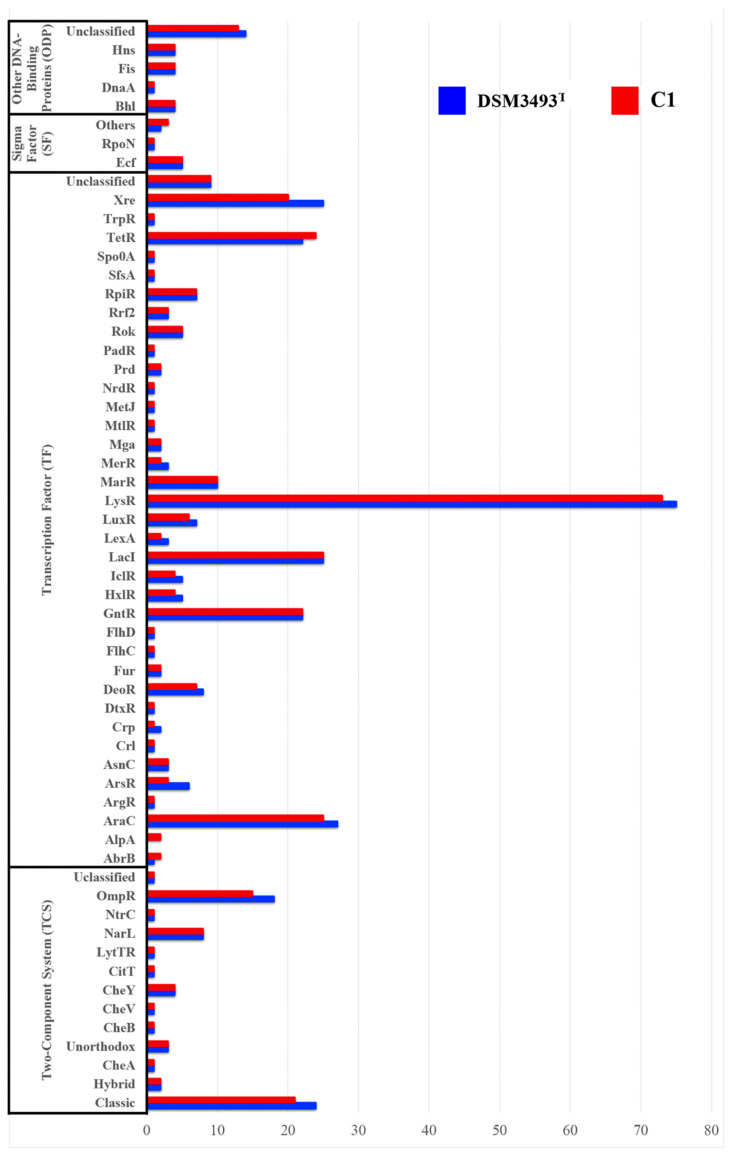
Classification and abundance of regulatory families in the genomes of *P. agglomerans* C1 (blue columns) and DSM 3497^T^ (red columns). The predictions were made using the P2TF database.

**Figure 9 microorganisms-13-02138-f009:**
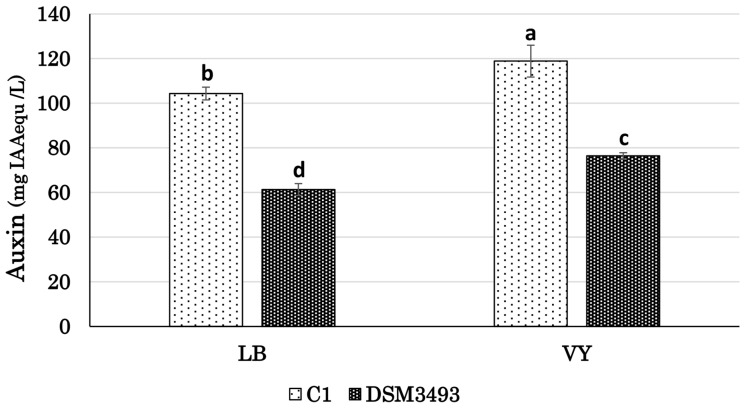
Effect of culture medium on auxin/IAA production (mg IAA_equ_ L^−1^) by *P. agglomerans* strain C1 and DSM3497^T^. LB and VY media contained the same amount of yeast extract (5 g/L) and differed in the source of peptone. Differences in letters indicate that the values are significantly different (*p* < 0.05).

**Table 1 microorganisms-13-02138-t001:** Metabolites whose abundance was significantly different (>2-fold; *p* ≤ 0.05) in the exometabolome of C1 vs. DSM3493^T^. The efs value indicates the ratio between the detector count values of C1 vs. DSM3493^T^. up = higher abundance in C1 exometabolome; down = higher abundance in DSM3493^T^ exometabolome. The growth rate in LB medium for both strains was comparable: 0.90 ± 0.12/h for strain C1 and 0.88 ± 0.16/ for DSM3493^T^.

**Compound_Name**	***p*-Value**	**Class**	**Edirection**	**efs**	**FDR**
Indole-3-ethanol (*Tryptophol*)	1.50 × 10^−7^	indoles	up	38	2.00 × 10^−5^
D-Fructose	2.40 × 10^−6^	hexoses	up	5.7	3.10 × 10^−4^
3-Indoleacetic acid	3.00 × 10^−7^	indoleacetic acids	up	3	4.00 × 10^−5^
Tyr-Pro	6.50 × 10^−9^	dipeptides	up	2.8	9.40 × 10^.7^
4-Methyl-5-thiazoleethanol	3.40 × 10^−5^	azoles	up	2.6	4.30 × 10^−3^
2′,3′-Dimethoxy-3-hydroxyflavone	3.70 × 10^−8^	chromones	up	2.4	5.30 × 10^−6^
Quinoline	2.00 × 10^−9^	quinolines	down	−2	2.90 × 10^−7^
Adenine	3.10 × 10^−10^	purines	down	−2.5	4.60 × 10^−8^
Deisopropylatrazine	1.60 × 10^−10^	triazines	down	−2.8	2.40 × 10^−8^
DL-Pyroglutamic acid (*5-Oxoproline*)	9.00 × 10^−12^	dipeptides	down	−3.1	1.40 × 10^−9^
Quinoline-2,8-diol	4.20 × 10^−10^	quinolones	down	−3.2	6.20 × 10^−8^
Adenosine	3.50 × 10^−9^	purines	down	−6.3	5.10 × 10^−7^
Hydrocortisone 17-acetate	4.60 × 10^−7^	methylprednisolone	down	−6.6	6.40 × 10^−6^
6-Methoxytryptamine	2.60 × 10^−7^	indoles	down	−11	3.50 × 10^−5^
Marrubiin	2.90 × 10^−9^	diterpenes	down	−19	4.20 × 10^−7^
1-Butylimidazole	6.50 × 10^−7^	imidazoles	down	−240	9.00 × 10^−6^
2-Methyl-4-nitroaniline	3.10 × 10^−3^	others	down	−440	3.80 × 10^−1^

## Data Availability

The original contributions presented in this study are included in the article/Appendix A. Further inquiries can be directed to the corresponding author.

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
