# Peer review of "Integrative Genomics and Metabolomics Analyses Provide New Insights into the Molecular Basis of Plant Growth Promotion by *Pantoea agglomerans"

_microorganisms, 2025, doi:10.3390/microorganisms13092138_

Round 1
Reviewer 1 Report
Comments and Suggestions for Authors
In this paper, the authors analyzed in silico 20 available genomes of the bacteria Pantoea agglomerans. The paper contains the results of the pan and core genomes of this group of bacteria with an emphasis on genes involved in IAA production, nitrogen and sulfur metabolism, defense, secretion, and oligopeptide transport. Furthermore, the authors compare two strains of the bacteria Pantoea C1strain and DSM3493T strain. The results of the comparison of the two strains related to bacteria plant and plant growth promotion are poorly documented and presented in the supplementary part of the article. It will be more interesting for the scientific community to see those data more detailed.
Figures 6 and Fig 7 are not necessary.
Figure 9 must be clear where limits between each group of proteins are.
Fig 10.Contains data for conversion of Trp in to IAA. Kinetics of Trp to IAA conversion is missing.
Table 1 is not clear. It is not clear under which conditions measurements were done. Two strains were grown in LB with Trp but no data about growth rate of two strains. Conclusions of production of different metabolites could be related to growth rate.
Finally, the title of the article must be in accordance with what is really described in the article, unfortunately data about biocontrol are missing.
Author Response
We thank the Reviewer for the valuable comments. The manuscript was revised following her/his suggestions.
General comment: The results of the comparison of the two strains related to bacteria plant and plant growth promotion are poorly documented and presented in the supplementary part of the article. It will be more interesting for the scientific community to see those data more detailed.
Response: The manuscript has been revised following the reviewer’s suggestions. The supplementary Figure S1 has been transferred to the main document, and the data are described in detail. More information about genes involved in plant growth promotion was presented in the revised section 3.3.
Comment: Figures 6 and Fig 7 are not necessary.
Response: Figures 6 and 7 have been removed from the revised manuscript and transferred to the Supplementary section as Figure S1 and Figure S2, respectively.
Comment: Figure 9 must be clear where limits between each group of proteins are.
Response: Figure 9 has been revised following the reviewer’s recommendation
Comment: Fig 10.Contains data for conversion of Trp in to IAA. Kinetics of Trp to IAA conversion is missing.
Response: The text was changed in the revised manuscript to provide additional data about the IAA productivity of the two strains. “Data obtained after 18 hours of incubation indicated that……”. Lines 425-488
Comment: Table 1 is not clear. It is not clear under which conditions measurements were done. Two strains were grown in LB with Trp but no data about growth rate of two strains.
Response: Data about the growth rate of the two strains has been added in the caption of Table 1 in the revised manuscript. “The growth rate in LB medium for both strains was comparable: 0.90 ± 0.12/h for strain C1 and 0.88 ± 0.16/h/ for DSM3493T”. Lines 454-458
Comment: the title of the article must be in accordance with what is really described in the article, unfortunately data about biocontrol are missing.
Response: The title of the manuscript has been modified as follows: “Integrative Genomics and Metabolomics Analyses Provide New Insights Into the Molecular Basis of Plant Growth Promotion by Pantoea agglomerans”.
Reviewer 2 Report
Comments and Suggestions for Authors
In the reviewed MS, the results of the comprehensive study on PGPR using genomic and metabolomic methods are presented. I read the paper with great interest. It is necessary to note that the quality of the study is very high. All parts of the MS are very detailed. The authors used proper methodology and statistical analysis. The obtained results are illustrated by high-quality figures. The Results and Discussion sections can be used as a standard of scientific research for other investigators, as they are very accurate and reflect all aspects of the research. The conclusions are based on the results and are robust. It should be emphasized that supplementary materials contain much useful data. Data availability statements are adequate.
Based on the importance and quality of the study, I recommend the MS for publication in the journal “Microorganisms” after some corrections.
Major suggestions:
- I recommend adding several recent (published in 2020-2025) references.
- In the last paragraph of the discussion (lines 697-700), it is better to move to the conclusions. Besides, in the conclusions you could discuss the importance of your study for science and technology in more detail.
Minor suggestions:
- Replace “Pantoea agglomerans” with another term, because you already mentioned it in the title. Correct “pangenome” to “pangenome analysis.” “Pantoea agglomerans C1” is unclear; call it, for example, “Pantoea agglomerans strain C1.”
- Line 35 and further: Correct [1; 2; 3] to [1-3].
- Line 112: Add reference for NCBI Taxonomy.
- Add information in sections 2.3, 2.4, and 2.10, or combine them with other sections.
- Figure 2: Enlarge the legend of the figure.
- Figure 3: Please, enlarge the figure.
- Line 326: P. agglomerans should be italicized.
- Figure 7: What do the colors blue, peach, and burgundy mean in the Venn diagram?
- Line 426: What does “LB medium” mean? Add it in “Abbreviations” at the end of the MS.
- Table 1: Please correct the table according to the journal requirements.
- Figure S1 supplementary: Please, increase the resolution of the labels.
Author Response
We thank the Reviewer for the valuable comments. The manuscript was revised following her/his suggestions.
Comment: I recommend adding several recent (published in 2020-2025) references.
Response: We introduced recent references through the revised manuscript, following the reviewer’s suggestion.
Comment: In the last paragraph of the discussion (lines 697-700), it is better to move to the conclusions. Besides, in the conclusions you could discuss the importance of your study for science and technology in more detail.
Response: Thanks for the suggestion. We moved the last part of the discussion to the conclusion paragraph, and details on the importance of the results obtained in this work have been included in the revised manuscript. “Furthermore, our results show that the integrated application of comparative genomics and metabolomics can be utilized to accurately predict the potential biotechnological applications….” Lines 715-725 in the revised manuscript.
Comment: Replace “Pantoea agglomerans” with another term, because you already mentioned it in the title. Correct “pangenome” to “pangenome analysis.” “Pantoea agglomerans C1” is unclear; call it, for example, “Pantoea agglomerans strain C1.” Response: In the keywords, Pantoea agglomerans has been changed to microbial biostimulants, and the pangenome has been corrected to pangenome analysis (in keywords and the revised manuscript). In the Keyword and all text, Pantoea agglomerans C1 has been changed to Pantoea agglomerans strain C1.
Comment: Line 35 and further: Correct [1; 2; 3] to [1-3].
Response: The references enumeration has been corrected as suggested by the Reviewer.
Comment: Line 112: Add reference for NCBI Taxonomy.
Response: References have been added to the revised manuscript.
Comment: Add information in sections 2.3, 2.4, and 2.10, or combine them with other sections.
Response: Information in sections 2.3 and 2.4 of the materials and methods has been combined with section 2.2 (Comparative and Pan-Genome Analysis), and information in section 2.10 has been combined with section 2.9 (Untargeted Metabolomics and Statistical Analysis) in the revised manuscript as suggested by the Reviewer.
Comment: Figure 2: Enlarge the legend of the figure.
Response: The legend of Figure 2 has been enlarged as suggested by the Reviewer.
Comment: Figure 3: Please, enlarge the figure.
Response: Figure 3 has been enlarged as suggested by the reviewer.
Comment: Line 326: P. agglomerans should be italicized.
Response: P. agglomerans has been italicized in the revised manuscript.
Comment: Figure 7: What do the colors blue, peach, and burgundy mean in the Venn diagram?
Response: The legend of the Figure has been changed to clarify the meaning of the colours in the Venn diagram. The figure has been moved to the Supplementary Material (Figure S2) as suggested by the other Reviewer.
Comment: Line 426: What does “LB medium” mean? Add it in “Abbreviations” at the end of the MS.
Response: The suggestion has been accepted, and the abbreviation has been added to the Abbreviation table.
Comment: Table 1: Please correct the table according to the journal requirements.
Response: As suggested by the reviewer, Table 1 has been corrected following the journal requirements.
Comment: Figure S1 supplementary: Please, increase the resolution of the labels.
Response: The resolution of the labels has been increased, and the Figure has been moved in the main manuscript, new Figure 6, following the suggestions of the other Reviewer.